

# The effects of spatial and temporal replicate sampling on eDNA metabarcoding

Kevin K. Beentjes[1,2], Arjen G. C. L. Speksnijder[1], Menno Schilthuizen[1,2], Marten Hoogeveen[1] and Berry B. van der Hoorn[1]

[1] Naturalis Biodiversity Center, Leiden, The Netherlands
[2] Institute of Biology Leiden, Leiden University, Leiden, The Netherlands

## ABSTRACT

**Background:** The heterogeneous nature of environmental DNA (eDNA) and its effects on species detection and community composition estimates has been highlighted in several studies in the past decades. Mostly in the context of spatial distribution over large areas, in fewer occasions looking at spatial distribution within a single body of water. Temporal variation of eDNA, similarly, has mostly been studied as seasonality, observing changes over large periods of time, and often only for small groups of organisms such as fish and amphibians.

**Methods:** We analyzed and compared small-scale spatial and temporal variation by sampling eDNA from two small, isolated dune lakes for 20 consecutive weeks. Metabarcoding was performed on the samples using generic COI primers. Molecular operational taxonomic unit (MOTUs) were used to assess dissimilarities between spatial and temporal replicates.

**Results:** Our results show large differences between samples taken within one lake at one point in time, but also expose the large differences between temporal replicates, even those taken only 1 week apart. Furthermore, between-site dissimilarities showed a linear correlation with time frame, indicating that between-site differences will be inflated when samples are taken over a period of time. We also assessed the effects of PCR replicates and processing strategies on general patterns of dissimilarity between samples. While more inclusive PCR replicate strategies lead to higher richness estimations, dissimilarity patterns between samples did not significantly change.

**Conclusions:** We conclude that the dissimilarity of temporal replicates at a 1 week interval is comparable to that of spatial replicate samples. It increases, however, for larger time intervals, which suggests that population turnover effects can be stronger than community heterogeneity. Spatial replicates alone may not be enough for optimal recovery of taxonomic diversity, and cross-comparisons of different locations are susceptible to inflated dissimilarities when performed over larger time intervals. Many of the observed MOTUs could be classified as either phyto- or zooplankton, two groups that have gained traction in recent years as potential novel bio-indicator species. Our results, however, indicate that these groups might be susceptible to large community shifts in relatively short periods of time, highlighting the need to take temporal variations into consideration when assessing their usability as water quality indicators.

Corresponding author
Kevin K. Beentjes,
Kevin.Beentjes@naturalis.nl

## INTRODUCTION

The importance of freshwater biodiversity and its effects on ecosystem resilience and stability have been well documented, and its monitoring is regulated by legislation such as the European Union Water Framework Directive of 2000 (EU WFD; Directive 2000/60/EC). Monitoring of biological quality elements (BQE), such as macroinvertebrates, is prescribed under the WFD, but traditional methods employed in this field are often considered slow, expensive, and sensitive to human-induced bias and errors (*Clarke & Hering, 2006*). Integration of molecular tools has been a focal area within this field of research for the past decade. The use of environmental DNA (eDNA) metabarcoding for species detection is gaining traction, as it would potentially enable to circumvent cumbersome traditional collection or visual observation of specimens. The use of eDNA for detection is based on the fact that organisms living in a certain environment, such as freshwater, leave behind traces of their existence via shedding and excretion of DNA. This technique has been applied successfully for the detection of a multitude of species, including BQEs, in both vertebrates (*Ficetola et al., 2008*; *Hänfling et al., 2016*; *Olds et al., 2016*) and invertebrates (*Thomsen et al., 2012*; *Schneider et al., 2016*; *Klymus, Marshall & Stepien, 2017*).

The heterogeneous nature of eDNA has been investigated in several model organisms, for example amphibians, where it was shown that spatial sampling increased the detection probability (*Dejean et al., 2012*; *Schmidt et al., 2013*). Similarly, richness estimates from eDNA community metabarcoding are sensitive to sampling strategies (*Grey et al., 2018*). This suggests that eDNA may only represent very local signals, especially in standing waters. It is therefore often recommended to include spatial coverage in an eDNA sampling strategy, either by sampling various points within a water body, or by combining all these samples into one large sample representing the entire water body (*Goldberg et al., 2016*; *Grey et al., 2018*; *Harper et al., 2019*). In addition to spatial sampling, temporal replicates may also increase detection probability, and provide a more complete impression of species richness and community composition. Many studies have examined the effects of spatial and temporal sampling on (macroinvertebrate) communities (*Baselga et al., 2013*; *Barsoum et al., 2019*), but limited work has been done on seasonal variation in aquatic eDNA. Most research focuses on one particular organism or groups of organisms, such as fish (*Stoeckle, Soboleva & Charlop-Powers, 2017*; *Sigsgaard et al., 2017*), amphibians (*Rees et al., 2017*; *Buxton, Groombridge & Griffiths, 2018*), and chironomids (*Bista et al., 2017*), or assesses the seasonal differences only at a limited number of points in time (*Chain et al., 2016*; *Guardiola et al., 2016*).

In this paper, we compare the effects of both spatial and temporal replicate sampling of eDNA within two isolated, but nearby, lakes, using a generic COI primer set. We assess patterns in communities based on molecular operational taxonomic unit (MOTU) clustering, identifying MOTUs using a lowest common ancestor (LCA) approach, and also

look at the communities of only those MOTUs identified as metazoans. Furthermore, we assess the impact of PCR replicates and subsequent sequence or bioinformatics processing strategies on the observed patterns of eDNA through space and time. We also highlight some potential opportunities and caveats in the use of eDNA for freshwater quality monitoring.

## MATERIALS AND METHODS

### Field sampling

Samples were collected on every Monday for 20 consecutive weeks, from May 2016 to September 2016, from two permanent lakes in a Natura 2000 protection area in the dunes of Wassenaar, the Netherlands. Two locations were selected, approximately 1.9 km apart: Location 1 "De Ezelenwei" (52.161°N, 4.354°E) and Location 2 "De Drie Landjes" (52.176°N, 4.367°E). The sampling window coincides with the sampling period for traditional WFD monitoring. Within each location three sub-sites were selected around the lake, roughly equidistant from each other (40–60 m apart) and representing different habitats and substrates. A total of 1 l of water was taken by submerging a 1-l sterile bottle slightly below the surface, one meter away from the lake shoreline. The bottles were brought back to the laboratory for filtration. As the sites were located in a nature conservation area, a permit was obtained from Staatsbosbeheer (2016/022).

### DNA filtration and extraction

Environmental DNA filtration was performed in the laboratory within 4 h after collecting the samples in the field. Sterilized Nalgene filter units (Thermo Fisher, Waltham, MA, USA) attached to a vacuum pump with 0.2 μm polyethersulfone filter membranes (Sartorius, Göttingen, Germany) were used to filter 300 ml of water. Filter holders were sterilized using 10% bleach solution and placed under UV-light for 30 min before use. After filtration, the filter membranes were stored in 900 μl CTAB buffer at −20 °C until extraction. DNA was extracted using a modified CTAB extraction protocol, adapted from *Turner et al. (2014)*. DNA precipitation was performed on 800 μl of aqueous phase, and final resuspension of the pellet was performed in 50 μl AE buffer (Qiagen, Venlo, the Netherlands).

### DNA amplification and MiSeq sequencing

A 316 bp fragment of the COI barcode region was amplified using primers BF1 and BR2 (*Elbrecht & Leese, 2017*). All sampling replicates were amplified in three independent PCRs, which were sequenced separately without pooling. A dual indexed MiSeq amplicon library was prepared using a two-step PCR protocol, in which the first PCR used primers BF1 and BR2 with 5′ Illumina tails (Tables S1 and S2). PCRs for round 1 were performed in 25 μl reactions containing 1× Qiagen CoralLoad PCR Buffer, 0.5 mM dNTPs, 0.05 U/μl Taq polymerase (Qiagen, Venlo, the Netherlands), 0.4 μM of each primer and 1.0 μl of template DNA. Initial denaturation was performed at 94 °C for 3 min, followed by 40 cycles at 94 °C for 15 s, 50 °C for 30 s, and 72 °C for 40 s, followed by final elongation at 72 °C for 5 min. Each 96-well plate contained blanks with no template

DNA and positive controls of Reeve's muntjac (*Muntiacus reevesi*) DNA extract to enable detection of cross-contaminations in the laboratory process. PCR success was checked on an E-Gel 96 pre-cast agarose gel (Thermo Fisher, Waltham, MA, USA). PCR products where then cleaned with a one-sided size selection using NucleoMag NGS-Beads (Macherey-Nagel, Düren, Germany), using a 1:0.9 ratio.

Second round PCRs were performed using 2.0 µl of PCR product from the first round in a 20 µl reaction containing $1\times$ TaqMan Environmental Master Mix 2.0 (Thermo Fisher, Waltham, MA, USA) and 1.0 µM of each primer. Initial denaturation was performed at 95 °C for 10 min, followed by 11 cycles at 95 °C for 30 s, 55 °C for 60 s, and 72 °C for 30 s, followed by final elongation at 72 °C for 7 min. Second round PCR products were quantified on the QIAxcel (Qiagen, Venlo, the Netherlands) and pooled equimolarly per PCR plate. Pools were cleaned with a one-sided size selection using NucleoMag NGS-Beads, ratio 1:0.9, then quantified on the Bioanalyzer 2100 (Agilent Technologies, Santa Clara, CA, USA) with the DNA High Sensitivity Kit. The four pools were combined equimolarly and sequenced on one run of Illumina MiSeq (v3 Kit, $2 \times 300$ paired-end) at LGTC (Leiden, the Netherlands).

## Quality filtering and MOTU clustering

Quality filtering and clustering of all data was performed in a custom pipeline on the OpenStack environment of Naturalis Biodiversity Center through a Galaxy instance (*Afgan et al., 2018*). Raw sequences were filtered using Sickle (*Joshi & Fass, 2011*) and merged using FLASH v1.2.11 (*Magoč & Salzberg, 2011*); all non-merged reads were discarded. Samples were split based on the presence of template-specific additional bases between Illumina tail and template-specific primers with a custom tool, and primers were trimmed from both ends of the merged reads using Cutadapt v1.16 (*Martin, 2011*). Any read without both primers present and anchored was removed. PRINSEQ v0.20.4 (*Schmieder & Edwards, 2011*) was used to filter reads with length below 310 bp and above 316 bp from the dataset. Sequences were dereplicated using VSEARCH v2.4.3 (*Rognes et al., 2016*) and clustered into MOTUs using UNOISE3 (*Edgar, 2016*) with an alpha of 0.5. The presence of *M. reevesi* reads in the non-control samples was used to determine the MOTU filtering threshold, only MOTUs with read abundances above 0.05% were retained for each replicate. Geneious 8.1.8 (https://www.geneious.com) was used to check for and remove MOTUs with indels and/or stop codons.

## Taxonomic assignment and diversity analysis

BLAST+ (*Camacho et al., 2009*) was used to compare MOTU sequences to a custom-made reference library containing COI sequences and bacterial genomes downloaded from NCBI GenBank (*Benson et al., 2005*) (sequences downloaded August 21, 2018). MEGAN v6.12.5 (*Huson et al., 2007*) was used to assign higher-rank taxonomy to MOTUs using the LCA approach from the top 100 hits from BLAST (settings: minimum bit score 170, minimum percent identity 80, top percent 5). The VEGAN package (*Oksanen et al., 2007*) in R was used to calculate beta diversity (Sørenson dissimilarity) between replicates and time points, make NMDS plots, and calculate correlations between dissimilarity

matrices and between the sample dissimilarity and sampling intervals. PCR replication effects were assessed using three methods of replicate processing: (1) counting all MOTUs toward the sample ("additive"), (2) only counting those MOTUs that appear in a majority of the samples ("relaxed"), or (3) only counting those MOTUs that occur in all replicates ("strict") (*Alberdi et al., 2018*). All analysis on the data were performed for both the whole dataset (all MOTUs), and a subset of the data with only metazoan MOTUs.

## RESULTS

### Sequencing run statistics

A total of 7,692,379 read pairs were obtained after sequencing. After merging and quality filtering, 5,743,638 sequences were retained for MOTU clustering. *M. reevesi* reads were detected in several non-control samples. Using a 0.05% threshold for filtering low-abundance MOTUs from each sample removed muntjac reads from all but one sample (Location 1.2, May 16). After filtering the MOTU table, 1,333 MOTUs were retained in the non-control samples. An additional 19 MOTUs with indels and stop codons were removed, resulting in a dataset with 1,314 MOTUs, representing 4,197,403 reads. Four samples with fewer than 2,000 reads were discarded. On average, PCR replicates had 11,790 reads (range 2,296–73,477), and 72 MOTUs (range 12–177). There was no correlation between number of reads and number of MOTUs in each sample.

### Taxonomic composition

Out of 1,314 remaining MOTUs, 530 (40.3%) eukaryotes could be identified to at least phylum level using the LCA, 119 (9.1%) were only classified as "eukaryote," 62 (4.7%) were identified as bacteria and 603 (45.9%) were not assigned any classification (Fig. 1). Within the eukaryotes, most MOTUs (318) were classified as stramenopiles. Of the 176 metazoans, 121 were identified as arthropods, mostly assigned to branchiopods (44 MOTUs) and insects (26 MOTUs). Of the 1,314 MOTUs, 537 (40,9%) were found in both lakes, 418 MOTUs were unique to location 1 (De Ezelenwei), and 359 MOTUs unique to location 2 (De Drie Landjes).

The MOTU communities differed significantly between the two lakes for all 20 sampling moments, which is reflected in the NMDS plot based on the Sørenson dissimilarity matrix (Fig. 2). Clustering of samples into their respective lakes was supported by ANOSIM ($R = 0.710$, $p = 0.001$). Similarly, ANOSIM also supported grouping of samples into two seasonal groups, spring (2 May–13 June), and summer (20 June–12 September) ($R = 0.486$, $p = 0.001$). For the metazoan-only subset, the separation between the locations is still supported by ANOSIM, albeit not as clear as in the dataset with all MOTUs ($R = 0.424$, $p = 0.001$). The grouping into spring and summer is also supported ($R = 0.587$, $p = 0.001$).

### PCR replicates

Out of 1,314 MOTUs, 110 only ever occurred in one PCR replicate, with an average of $14.0 \pm 1.6$ (mean ± SEM) reads. The other 1,204 MOTUs occurred on average in $21.2 \pm 1.1$ of the 356 total replicates. No MOTU was found in all replicates. Average Sørenson dissimilarity between PCR replicates was 0.26 (Fig. 3). Using the "additive" PCR processing strategy, samples had an average of $102.5 \pm 4.0$ MOTUs. Under the "relaxed"

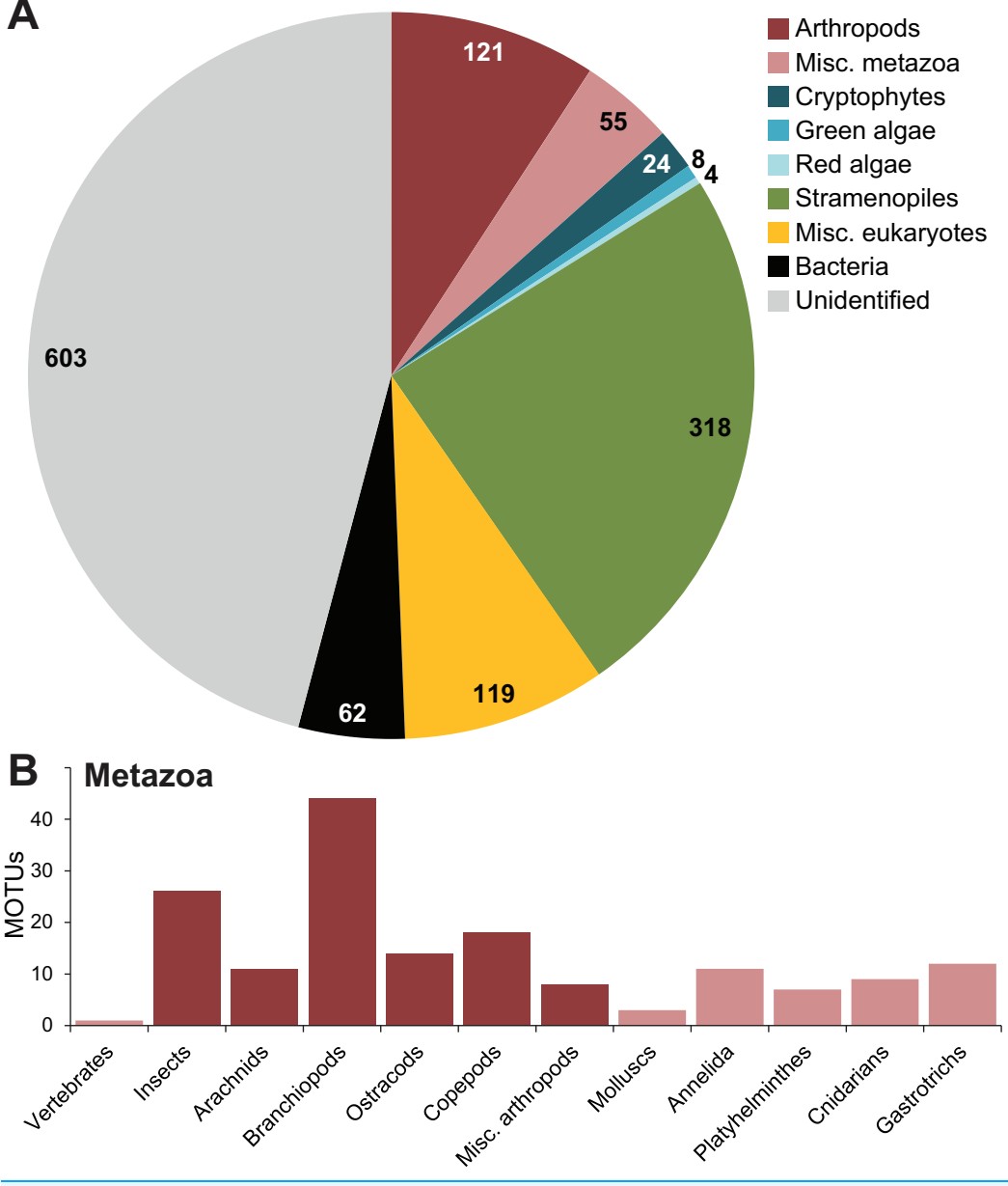

**Figure 1  Taxonomic assignment of MOTUs.** Taxonomic assignments of the MOTUs at (A) phylum-level and (B) class-level for metazoa, using a lowest common ancestor approach in MEGAN. Numbers in the pie chart indicate the number of MOTUs assigned to each phylum.

scenario samples had an average of 65.7 ± 2.4 MOTUs, and 280 MOTUs were discarded from the MOTU table. In the "strict" scenario an additional 246 MOTUs were discarded (Table 1). The remaining 788 MOTUs still represented 95.1% of the total read data. One PCR replicate on average contained 70.9% of MOTUs found in the total spatial replicate sample (the three PCR replicates combined) (range 34.6–95.8%), two replicates combined were able to detect an average of 88.4% of the MOTUs (range 55.8–100%). In only ten of 120 samples, the addition of a third PCR replicate did not result in additional

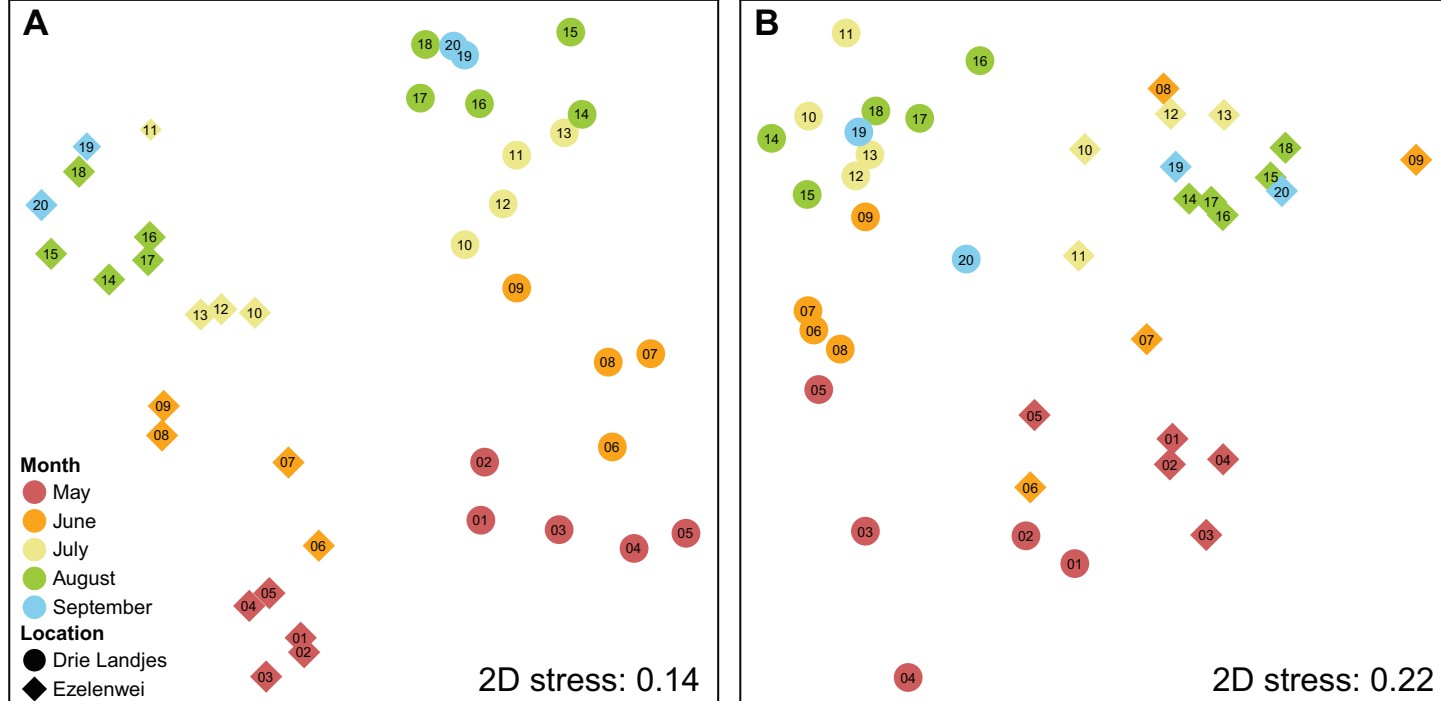

**Figure 2 NMDS plots representing the dissimilarities between sampling sites.** Two-dimensional NMDS plots constructed based on Sørenson dissimilarities between sampling sites using (A) all MOTUs and (B) only metazoan MOTUs. 2D stress values are displayed in the panels. Each point represents the combined community of the three spatial sampling replicates taken at each of the two locations on each of the 20 time points, with the PCR replicates combined using the "additive" strategy. Shapes indicate the location, colors are used to indicate the month in which samples were obtained, with numbers labeling the consecutive weeks from 2 May to 12 September. ANOSIM supported grouping of the samples belonging to one lake for both all MOTUs as the metazoan-only dataset ($R = 0.710$ and $R = 0.424$, respectively, $p = 0.001$). Seasonal grouping was similarly supported by ANOSIM, splitting samples into two seasonal groups (2 May–13 June, and 20 June–12 September) ($R = 0.486$ and $R = 0.587$, respectively, $p = 0.001$) for all MOTUs and metazoan-only.

MOTUs found. Seven of the PCR replicates contained no MOTUs that could be identified as metazoan, two subsamples had no metazoan MOTUs in any of their PCR replicates. Average Sørenson dissimilarity between PCR replicates in the metazoan-only subset of the data was 0.18 (Fig. 3), although in some cases it was as high as 1.0.

## Sampling replicates

Average Sørenson dissimilarity between sampling replicates within one location at the same time point was 0.48 using the "additive" PCR replicate strategy (Fig. 3), and significantly higher than dissimilarities between PCR replicates (*t*-test, $p = 0.005$). When using the "relaxed" and "strict" approaches, the average was slightly lower (0.45 and 0.46, respectively) (Table 1), but not significantly different (ANOVA). Four samples with only two successful PCR replicates were omitted from this analysis. There was a strong correlation between the Sørenson dissimilarity matrices for sample replicates under all three PCR replicate processing strategies (Fig. S1), both for the dissimilarities between sampling replicates pairs, and the dissimilarity matrix as a whole.

The high dissimilarity between sampling replicates was reflected in the contribution of each sampling replicate to the total diversity of the lakes at each time point. The three

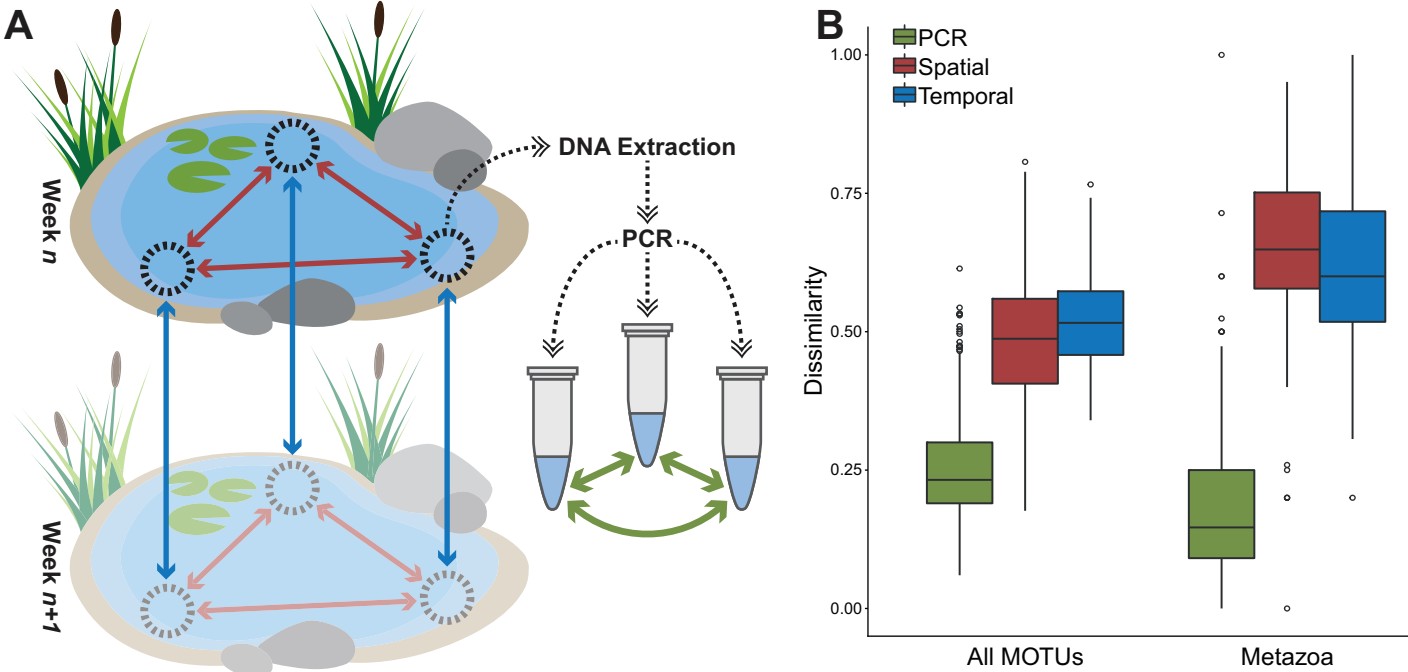

**Figure 3 Average dissimilarities between PCR replicates, spatial replicates, and temporal replicates.** (A) Schematic representation of the replicate sampling strategies and (B) boxplot displaying Sørenson dissimilarity values for PCR replicates (green, $n = 352$ for all MOTUs, $n = 344$ for metazoa), spatial sampling replicates (red, $n = 104$) and temporal replicates separated by 1 week (blue, $n = 100$) for both all MOTUs and metazoan-only MOTUs. In both cases, the dissimilarity between spatial replicates was significantly higher than between PCR replicates ($t$-test, $p = 0.005$). Only in the case of all MOTUs was the temporal dissimilarity significantly higher than the spatial dissimilarity ($t$-test, $p = 0.005$). There was no significant difference between spatial and temporal dissimilarities in the metazoan-only for samples taken 1 week apart.

**Table 1 Sample richness and heterogeneity under different replicate processing strategies.**

| MOTUs | PCR strategy | Richness | | Sample replicates | | | |
| --- | --- | --- | --- | --- | --- | --- | --- |
| | | **Total** | **Average** | **Common (3/3)** | **Shared (2/3)** | **Unique (1/3)** | **Dissimilarity** |
| All | Additive (1/3) | 1,314 | 187.3 ± 9.1 | 41.6 (22.2%) | 37.5 (20.0%) | 108.2 (57.8%) | 0.48 ± 0.01 |
| | Relaxed (2/3) | 1,034 | 114.8 ± 5.8 | 29.3 (25.5%) | 23.8 (20.7%) | 61.5 (53.8%) | 0.45 ± 0.01 |
| | Strict (3/3) | 788 | 81.8 ± 4.4 | 17.6 (21.5%) | 18.8 (22.9%) | 45.5 (55.6%) | 0.46 ± 0.01 |
| Metazoan | Additive (1/3) | 176 | 25.0 ± 2.4 | 18.0 (75.2%) | 4.8 (17.5%) | 2.3 (10.9%) | 0.65 ± 0.01 |
| | Relaxed (2/3) | 156 | 19.2 ± 2.3 | 14.2 (72.2%) | 3.8 (17.7%) | 1.7 (10.1%) | 0.66 ± 0.02 |
| | Strict (3/3) | 141 | 15.8 ± 2.2 | 12.0 (75.2%) | 2.6 (17.2%) | 1.1 (7.6%) | 0.68 ± 0.02 |

**Note:**
Total richness of all samples combined, as well as average (mean ± SEM) richness for each of the two locations at each of the 20 time points under different PCR replicate processing strategies ("additive," "relaxed," and "strict"), the effects on heterogeneity of MOTUs in the sample replicates and the average Sørenson dissimilarities between the sampling replicates (mean ± SEM). For each of the three strategies, the MOTUs are divided into three categories: (1) those MOTUs that are common and appear in all three sampling replicates; (2) MOTUs that are shared, and occur in two of three replicates; and (3) unique MOTUs, that only occur in a single sample replicate.

sampling replicates combined had an average of 187.3 ± 9.1 MOTUs, whereas a combination of two replicates only represented 81.0% (range 34.0–100%) of that total. In only one of 40 (two lakes, 20 time points) cases, the addition of a third sampling replicate did not provide additional MOTUs. One sampling replicate on average only produced 103.0 ± 3.9 MOTUs, which represented 55.4% of the total (range 12.1–92.5%). Regardless

of the PCR replicate processing strategy used, the average proportion of MOTUs unique to one of three sample replicates was roughly the same (Table 1).

## Temporal replicates

To look at the temporal patterns in the data, we used the "additive" PCR processing strategy, and added each of the three spatial replicates per week per location into one data point. This resulted in 40 data points with an average of 187.3 ± 9.1 MOTUs for 104,935 ± 5,007 reads. Again, there was no correlation between number of reads and number of MOTUs. A total of 257 (19.6%) MOTUs only ever occur in a single time point in a single location, only four MOTUs occur every week in both locations. Weekly samples represented between 9.5% and 37.9% (average 20.2%) of the total MOTU community observed in the lake, with later weeks generally having a higher richness than the earlier weeks. Turnover was not calculated as it was inflated by MOTUs occurring in non-consecutive weeks.

The average Sørenson dissimilarity between two replicates taken 1 week apart at the same sampling point was 0.53, which is significantly higher than the dissimilarity between two replicates taken at the same time (t-test, $p = 0.005$) (Fig. 3). With the sampling replicates combined, the Sørenson dissimilarity between the total communities of one location a week apart was 0.48 on average. Looking at larger time intervals, there was a significant correlation between interval duration and Sørenson dissimilarity (Spearman correlation $\rho = 0.812$, $p < 0.001$) (Fig. 4).

For the metazoan-only subset, dissimilarity between the sampling replicates and the temporal replicates was much higher than for the whole dataset, at 0.65 and 0.62, but with no significant difference between them (Fig. 3). Temporal replicates were significantly more dissimilar than spatial replicates for intervals of three or more weeks (t-test, $p = 0.002$). The same effects as with all MOTUs were seen when looking at the PCR replicate processing strategies, where average dissimilarities were not significantly different for each of the three strategies, albeit much higher than when using all MOTUs (Table 1). The correlation between interval duration and Sørenson dissimilarity was also significant for metazoan-only data (Spearman correlation $\rho = 0.555$, $p < 0.001$) (Fig. 4).

## DISCUSSION

Our results demonstrate the relatively large differences that can exist between sampling replicates, both on a spatial and a temporal scale. A significant challenge in the use of eDNA for metabarcoding stems from the heterogeneity of eDNA within the environment, and also in DNA extracts. The latter introduces a stochastic effect when sequencing multiple PCR replicates, in which less abundant species may not be found in all replicates. We applied three ways of bioinformatics processing of PCR replicates: (1) using all MOTUs ("additive"), (2) only using MOTUs present in two or more replicates ("relaxed"), and (3) only using MOTUs present in all three replicates ("strict") (Alberdi et al., 2018). Whilst the chosen strategy had an impact on the total and average number of MOTUs found in each sample, general patterns of dissimilarities between samples were not largely impacted.

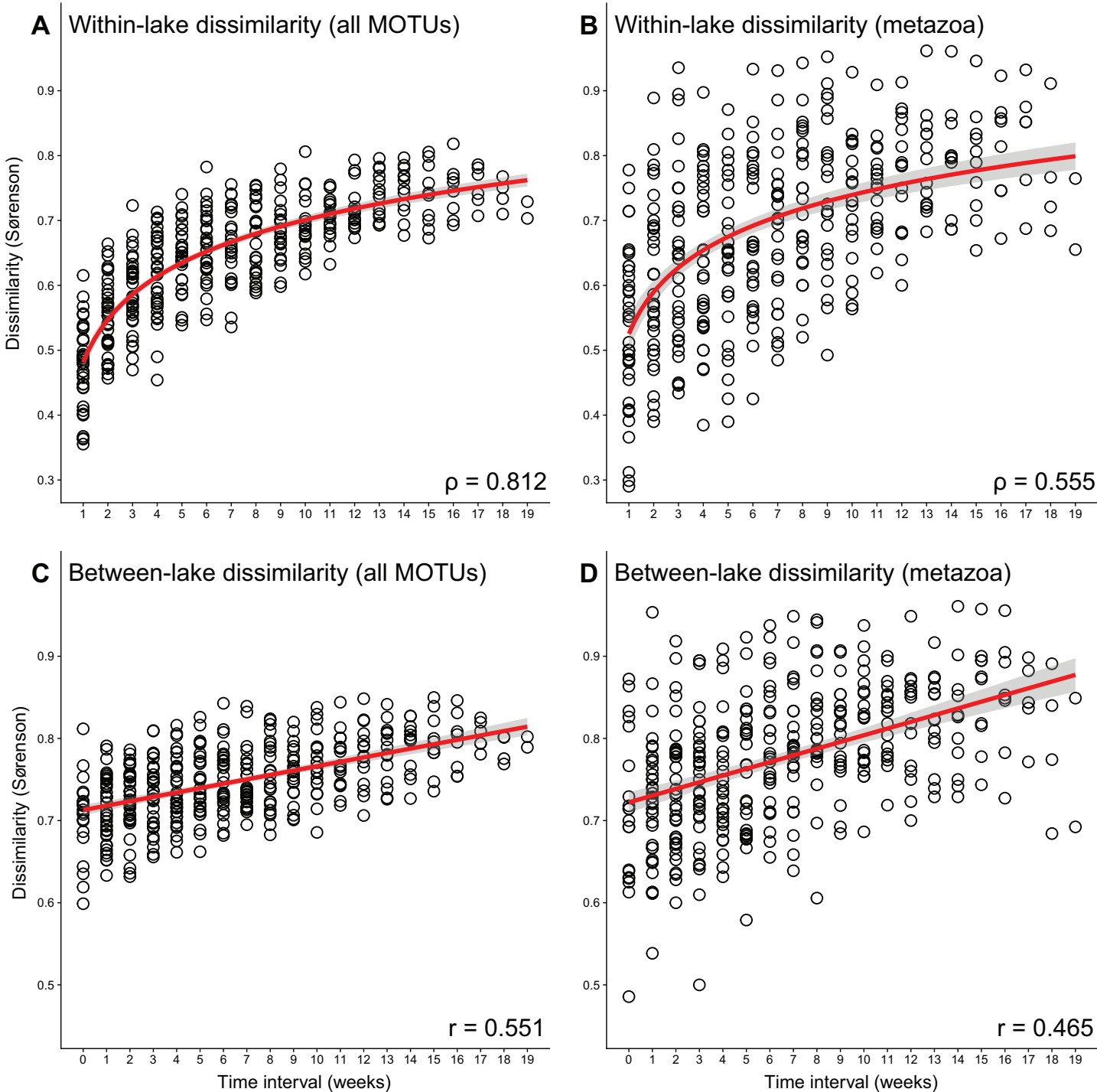

**Figure 4 Correlations between time interval and sample dissimilarity.** Time interval between two sampling moments vs. the Sørenson dissimilarity between total communities for samples taken in the same lake, with (A) all MOTUs and (B) only metazoan MOTUs (Spearman correlation, *p* < 0.001), and time interval between two sampling moments vs. the Sørenson dissimilarity between total communities for samples taken in different lakes, with (C) all MOTUs, and (D) only metazoan MOTUs (Pearson correlation, *p* < 0.001) (with 95% confidence interval). Correlation values are provided in the panels. Sampling replicates are merged into one sample per location per week, PCR replicates are processed using the "additive" strategy.
When we look at the heterogeneity of eDNA across the three sampling replicates within one location at a given time, the proportion of MOTUs that occur in either one or in all of the samples stays the same regardless of PCR replicate processing strategy. This indicates that removal of MOTUs not covered by all PCR replicates (the "strict" strategy) does not necessarily make spatial replicates more similar. This observation is confirmed by the average dissimilarity between the spatial samples, which is not significantly different for any of the three PCR replicate strategies (Table 1). Similarly, the Sørenson dissimilarity matrices were highly correlated ($r = 0.929$ and $r = 0.917$ for "additive" vs. "relaxed" and "relaxed" vs. "strict," respectively. Pearson correlation, $p < 0.001$) (Fig. S1). This suggests that the selected strategy can vary depending on the research question without significantly impacting observed patterns of biodiversity, although it affects the richness estimates. PCR results are not always reproducible, as witnessed by the average dissimilarity of 0.26 between PCR replicates in this study, but also as reported in the detection of rare species (*Ficetola et al., 2008*; *Buxton, Groombridge & Griffiths, 2018*). Especially when looking for rare species, multiple PCR replicates improve detection chances. For analyses that benefit from more complete taxa lists, such as those performed for WFD monitoring, the inclusion of multiple PCR replicates also seems beneficial. While we only took three sampling replicates within each lake in each week, others have suggested as much as nine samples to estimate biodiversity from eDNA (*Grey et al., 2018*).

Compared to PCR replicates, the Sørenson dissimilarity between spatial replicates (0.48 on average for the full dataset, 0.65 for the metazoan-only subset) is significantly higher (Fig. 3), which reflects the heterogeneity of eDNA within the environment. Previous studies have already pointed out that eDNA signal can have strong local effects (*Moyer et al., 2014*; *O'Donnell et al., 2017*; *Stewart et al., 2017*), due to limited dispersal and sedimentation, but also the rapid degradation of eDNA (*Dejean et al., 2011*; *Barnes & Turner, 2015*). The use of spatial replicate sampling to retrieve eDNA results that are representative for the whole body of water has been stressed (*Goldberg et al., 2016*; *Harper et al., 2019*), and shown to improve eDNA monitoring efficacy (*Goldberg, Strickler & Fremier, 2018*). Resampling at different time points, however, has received little attention. Up until now research into seasonal variation has often focused on a limited set of temporal samples, such as spring vs. autumn/winter (*Chain et al., 2016*; *Guardiola et al., 2016*; *Lacoursière-Roussel et al., 2018*). The effects of temporal replicate sampling in this study were comparable with those of spatial replicates, with dissimilarities between samples taken at one sampling point a week apart slightly but significantly higher than those between samples taken within one lake at a certain week (average 0.53 vs. 0.48) (Fig. 3). Almost a fifth (19.6%) of MOTUs was only ever detected in a single time point. In the metazoan-only subset the spatial and temporal dissimilarities were higher than for the complete dataset (0.65 and 0.62, respectively), although not significantly different from each other. Temporal dissimilarity was significantly higher than spatial dissimilarity, however, for intervals of 3 weeks or more. Similar observations were made for example in fish (*Stoeckle, Soboleva & Charlop-Powers, 2017*; *Sigsgaard et al., 2017*), where many species were only detected in a few time points, showing that temporal sampling regimes are needed for optimal recovery of the total biodiversity. Our sampling time frame

coincides with the period in which most of the traditional WFD monitoring is performed, for which insights into within-season community changes are more relevant than between-season variations.

The data included a number of MOTUs occurring in non-consecutive weeks, suggesting these MOTUs went undetected, rather than being absent from the environment. A detection/non-detection cannot be directly translated into presence/absence (*Roussel et al., 2015*). These irregular patterns of occurrence may have increased the dissimilarity between replicate samples, both temporal and spatial. However, we observed a strong correlation between time interval and Sørenson dissimilarity (Spearman correlation, $\rho = 0.812$, $p < 0.001$) (Fig. 4). Interestingly, it is not a linear correlation, and there seems to be a maximum to the dissimilarity between samples taken at different time points. Although we only sampled for 20 consecutive weeks, this data suggest that the community never changes completely within this time frame. The maximum observed Sørenson dissimilarity between two samples taken at one sampling point is 0.90 (for a 9 week interval). This indicates that, even though there are large changes in eDNA composition between different time points, there is some basal community that is present throughout the sampling period and does not change. Such basal communities could be relevant for identifying potential novel targets for eDNA-based monitoring, as it would allow for a time-independent assessment. Planktonic crustaceans, such as the copepods and branchiopods found in relatively large numbers (both MOTUs and reads, Figs. S2 and S3) have the potential to be such new bio-indicators, as they may be more easily detected using eDNA and likely to respond quicker to environmental changes (*Lim et al., 2016*; *Montagud et al., 2018*). Additionally, we observed a linear increase in dissimilarity between the two locations over time (Pearson correlation, $r = 0.551$, $p < 0.001$). Average Sørenson dissimilarity of the two lakes was 0.71 when sampled in the same week (interval = 0), and increased up to 0.80 when sampled 19 weeks apart (Fig. 4). This indicates that studies comparing communities between locations should be wary of the time intervals between sampling, as larger intervals between sampling may lead to inflated dissimilarities.

Even though there are large differences between communities along the temporal gradient, there were no large shifts in the taxonomic compositions defined by LCA (Figs. S2 and S3). Other than an increase in the number of metazoan taxa over time (both in absolute number of MOTUs and in proportion of the total diversity), the proportional contribution of each of the different taxonomic groups is roughly the same for all 20 weeks, in both lakes. This indicates that seasonal succession mostly occurs within the taxonomic groups. The increase in metazoan taxa may be slightly inflated in the data for location 2, where algae (two MOTUs classified as Chrysophyceae) dominated the reads between 30 May and 20 June, and potently out competed others in both DNA extraction and amplification. The rest of the weeks in location 2, and all weeks in location 1 were mostly dominated by arthropod (copepod and branchiopod) and unidentified reads (average of 36.0% and 48.7%, respectively).

The primers used in this study perform well on macroinvertebrate bulk samples, but are degenerate enough to amplify a wide range of non-target DNA from non-metazoan

sources present in environmental samples that would normally not be found in bulk macroinvertebrate samples (Fig. 1). In our case, only 13.4% of the MOTUs could be assigned to metazoan phyla. Within those, only about a third (51 out of 176) could be assigned to phyla that are actually counted as macroinvertebrates for the purpose of traditional quality monitoring under the WFD. The remainder of the metazoans were mainly branchiopods and copepods. Similar results with non-target taxa were reported in other papers using degenerate COI primers for freshwater community metabarcoding (*Weigand & Macher, 2018*). There has been some debate about the usability of the standard COI barcode region defined by *Hebert et al. (2003)* within DNA- and eDNA-based analyses, but thus far the benefit of an extensive COI database seems to outweigh the drawbacks (*Andújar et al., 2018*), as also witnessed by the many primer sets that have been designed for macroinvertebrate metabarcoding studies (*Leray et al., 2013*; *Bista et al., 2017*; *Elbrecht & Leese, 2017*). The balance between universality of primers and target specificity is a delicate one, and metabarcoding "by-catch" can represent a significant share of the data. In our data, one fourth of the MOTUs were classified as stramenopiles and various algae groups. The COI barcode region may not be the optimal marker for all of these groups. Even in situations where not all MOTUs can be identified up to species level, unidentified (or partially identified, in the form of higher taxa) MOTUs can still be matched across different samples and may therefore still be of use for community analyses (*Lim et al., 2016*).

The primer sets used in this study may not have been optimal for recovery of all taxon groups, and group-specific primers may be more appropriate for the detection of novel bio-indicators. Nonetheless, we expect the temporal effects observed in this study to play a role in any community analysis. Even when eDNA is used for BQE monitoring, time intervals between sampling sites will likely remain, as it practically impossible to sample and process all sites within a short time frame. Seasonal effects have been reported in the rich history of publications based on morphological observation of seasonality in planktonic organisms (*Gosselain, Descy & Everbecq, 1994*; *Wu et al., 2013*), but molecular tools will allow for much finer resolution observations. We strongly encourage any research into the use of novel indicator taxa to take these temporal changes into consideration, as they clearly affect non-macroinvertebrate taxa such as the phyto- and zooplankton groups observed in this study.

## CONCLUSIONS

We here present the first study that directly compares the effects of small-scale spatial and temporal resampling eDNA for metabarcoding. We show that replication leads to better estimations of total biodiversity, where the effects of spatiotemporal sampling replicates are significantly greater than PCR replications, even though the latter can already bring a substantial increase in richness depending on the replicate processing strategy. Interestingly, the PCR replicate handling strategy has little effect on patterns in biodiversity and dissimilarity between samples, and there are no severe drawbacks of including even those MOTUs that occur in only one replicate. Dissimilarities between temporally separated samples were approximately equivalent to the dissimilarities

between spatially separated samples. These dissimilarities increase over longer time intervals, suggesting that population turnover effects are stronger than community heterogeneity. This is an important consideration for any study comparing multiple communities that have been sampled at different time points, as well as any study that delves into the use of novel bio-indicators. Non-macroinvertebrate taxa, such as the phyto- and zooplankton groups observed in this study, are often put forward as potential bio-indicators. The effects of sampling strategies, especially short-term temporal replicate sampling, can have a considerate impact on the usability of these taxa.

## ACKNOWLEDGEMENTS

We thank Staatsbosbeheer and Casper Zuyderduyn for access to the dune lakes and permits for field work. We also thank Dr. Oscar Vorst for his help in assessing potential sampling sites, Jeroen Visser for his assistance in the sample collection and laboratory work, and Elza Duijm for her assistance with NGS library preparations.

### Funding

This study funded by the Gieskes-Strijbis Fonds, as a part of the DNA Waterscan project. There was no additional external funding received for this study. The funders had no role in study design, data collection and analysis, decision to publish, or preparation of the manuscript.

### Grant Disclosures

The following grant information was disclosed by the authors:
Gieskes-Strijbis Fonds, as a part of the DNA Waterscan project.

### Competing Interests

The authors declare that they have no competing interests.

### Author Contributions

- Kevin K. Beentjes conceived and designed the experiments, performed the experiments, analyzed the data, prepared figures and/or tables, authored or reviewed drafts of the paper, approved the final draft.
- Arjen G. C. L. Speksnijder conceived and designed the experiments, authored or reviewed drafts of the paper, approved the final draft.
- Menno Schilthuizen conceived and designed the experiments, authored or reviewed drafts of the paper, approved the final draft.
- Marten Hoogeveen analyzed the data, contributed reagents/materials/analysis tools, approved the final draft.
- Berry B. van der Hoorn conceived and designed the experiments, authored or reviewed drafts of the paper, approved the final draft.

## Field Study Permissions

The following information was supplied relating to field study approvals (i.e., approving body and any reference numbers):

Fieldwork permit was obtained from Staatsbosbeheer (permit number 2016/022).

## Data Availability

All raw sequence data is available from the Short Read Archive (SRA): PRJNA529573.

## Supplemental Information

Supplemental information for this article can be found online at http://dx.doi.org/10.7717/peerj.7335#supplemental-information.

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
