# Peer review of "The effects of spatial and temporal replicate sampling on eDNA metabarcoding"

_PeerJ, doi:10.7717/peerj.7335_

## Round 0.1 · original submission · Major Revisions

Although both reviewers were enthusiastic about your study, they identified a number of issues that need to be addressed before it can be accepted.

·

Basic reporting

The ms meets all the requirements of basic reporting.

Experimental design

The ms meets all the requirements of experimental design.

Validity of the findings

My only major concern is that the relative magnitude of temporal heterogeneity is somewhat overstated. For example, in Line 32 (Abstract - Conclusions) they state “We conclude that temporal replicate samples were generally more dissimilar than spatial replicate samples, especially over longer time intervals...”. However, their analysis showed that temporal replicates at 1 week intervals were only significantly more dissimilar than spatial replicates if all MOTUs were considered (there was no significance if only considering metazoan MOTUs). Also, even the significant difference is not of great magnitude (Figure 3 “All MOTUs”). I think it would be better to conclude something more modest, such as “We conclude that the dissimilarity of temporal replicates at one-week intervals is roughly equivalent to that of spatial replicates, but that it becomes much larger as time interval increases.” I might also do an analysis to see at what interval the temporal replicates have significantly higher dissimilarity than spatial replicates in the metazoan dataset.

Additional comments

The manuscript “The effects of spatial and temporal replicate sampling on eDNA metabarcoding” estimates PCR, spatial, and temporal heterogeneity of community composition in two dune lakes. The authors concluce that there were relatively larger differences between temporal replicates than spatial replicates and these differences grew over time. This result suggests that it is very important to consider temporal scale when designing or interpreting eDNA metabarcoding studies. Heterogeniety between PCR replicates was present, but was relatively smaller than either temporal or spatial heterogeneity and did not influence dissimilarity patterns.

This is an important and interesting study that increases our understanding of the heterogeneity in freshwater phytoplankton and zooplankton eDNA communities and informs eDNA metabarcoding applications. The paper is well written, clear, and includes the appropriate citations and context. Besides the one major issue I have describe above (re over-stating the temporal dissimilarity somewhat), I only have a few minor comments/suggestions:

Minor Comments/Suggestions
Line 50 – delete “Especially”

Line 62 - “suggests”

Line 64 – delete “and creating many”

Line 92 – How were the subsites chosen? About how far apart were the subsites?

Methods question – Were any field blanks taken? What steps were taken to prevent cross-contamination between subsites and sites?

Line 294 – add temporal eDNA citation Lacoursière‐Roussel, Anaïs, et al. "eDNA metabarcoding as a new surveillance approach for coastal Arctic biodiversity." Ecology and Evolution 8.16 (2018): 7763-7777.

Figure 1 Legend. Please add the y-axis metric and explain the pie chart width metric.

Figure 4 Legend. Spearman rho is given in plots A and B but r is given in C and D – is this a typo or does it reflect a different analysis? Also, explain the error around the prediction lines.

·

Basic reporting

1.1 Article is well structured and written. Some sections lack a bit of clarity due to wording and missing information, but it's a minor issue.

1.2 Environmental surveying context, sampling issues and motivation are very well referenced. Methods are unclear at some points (sampling site, PCR and sequencing design and bioinformatic pipelines). Data is well explored and all major results and method biases are discussed with plenty of relevant references.

1.3 Structure follows guidelines. Figures, tables and supplementary materials are all according to text. Figures S2 and S3 are blurry. Table S2 could include the tagging so experimental design could be better evaluated.

1.4 Proposed research was tested and discussed. No deviations from main concept.

Experimental design

2.1 Research subject is within biological sciences (molecular ecology)

2.2 Question is well defined and relevant. Original sample design includes questions (PCR replicates) already answered in literature, but time correlation is the main point of the study and is well answered.

2.3 Experimental design lacks in clarity and controls.

A few notes:

Lines 91-93
Sampling sites were chosen at random every time or sampling occured in the same spots each week? It's not clear in the text or figures.
What were the distances between the sampling sites? Were the samples collected always in the same order? If the sampling sites are too close, collecting water in spot 1 can influence spot 2 and 3 results due to water movement and the collector itself carrying residual eDNA. Did the authors control for this bias?

Lines 112-114
Assay should be specified as final concentration instead of volumes.
* Did the assay use 0.05 nM of dNTPs for a 40 cycle PCR? There should be almost no amplification with this little dNTP amount.
* Used Taq polymerase should be specified as units instead of volume.
* Neither primer stock/ aliquot concentration or final assay concentration are specified (same for line 124)
* Sample concentration range should also be noted. Same issue for line 122. Why was the volume fixed instead of concentration?

PCR plate designs are not shown, so the cross-contamination controls cannot be assessed. Weeks are divided in 4 groups, each containing 5 sampling weeks (1-5,6-10,11-15 and 16-20). Each group uses the same inline barcode primers (BF/BR). This can lead to amplicons jumping within groups undetected if plates were designed poorly. Possible effects are: (a) similarity between groups will be overall smaller than it should, (b) MOTUs will jump betweeen locations and weeks as false detections and (c) PCR replicates for each sub-sampling site can show the same diversity, even if the real eDNA distribution is different.

Barcode combination design were optimized to split locations, PCR replicates and weeks within groups? Since chimeras removing parameters in UNOISE3 were not specified, a bad filtering can lead to MOTUs jumping between the ingroups (and even between groups if the filtering fails to detect inline indexes).

PCR duplicate is a well known technical artifact that increases the abundance of a sequence over others randomly. In metabarcoding context, it can lead to random MOTUs dominating over others. Usually, to avoid duplicates, samples (and/ or indexes) are marked with Unique Molecular Identifiers. This allows the identification of over amplified UMIs by random and removal by bioinformatics. In case UMIs are not used by lab design, PCR duplicates should still be assessed (using tools like Clumpify/ Picard/ SAMTools) and both data sets (with and without filtering) compared.

Why did alpha was set to 0.5 for UNOISE3? Any previous analysis? Setting the alpha this low for a primer that inclusive is probably generating too many MOTUs that belong to the same species. In this case, ZOTUs should be used. If alpha comparisons for this data set were ran, they should be presented in supplementar material.

Lines 168-169 Why did the authors keep cross-contamined wells instead of removing it completely?

Lines 171-172 Why were the MOTUs with stop codons or frame shifts removed? Degenerated primers (such as used) are known to cause polymerase slips and increase sequencing bases by a few pairs (Elbretch et al. 2018).

Is it possible that algae dominated the readings due to filter clogging? Manajeva et al. (2018) shows that small pore size favors sequencing of algae and bacteria, and that in this case, the samples should be pre filtered. Since PES is a surface filter, did the authors control for clogging (line 355)?
Wilcox et al. (2015) also shows that eDNA particle size is a key point for survey design. He suggests that bigger pores (up to around 1 micrometre) increase the filtering of animal eDNA particles as it doesn't retains smaller unicellular organisms. How the authors predict temporal correlactional will respond to changing eDNA particle size?

This one is just a small comment

Lines 257-260 PCR replicates from the same DNA extract being heterogeneous is a amplification (by primer design) and sequencing bias. Different DNA extracts for the sample sample yielding different MOTUs is a extraction method bias. Currently, there are several studies comparing eDNA extraction methods and their results, but there’s no consensus on which is the best. Sampling effect should also be noted from DNA extraction effect, as the sampling methodology is also not optimized.

Validity of the findings

3.1 Study shows 2 contributions to metabarcoding and environmental DNA area:
1) Temporal replicates are shown to be an important part of a metabarcoding surveilance design
2) Communities have basal MOTUs that are constant and can be used as control for other experiments, or as biological indicators

3.2 Data should be reanalized to reduce noise according to 2.3. DNA sequences are not deposited yet.

3.3 Study is limited to the proposed question. Ecological statistics are well discussed, but genetics need to be refined (which leads to new analysis). Text is well written and fundamented by results.

3.4
Suggestion and speculation:

The irregular pattern of occuring (lines 307-311) is very interesting. Due to described stochastic nature of PCR, it could be possible that the original DNA extracts has the original sequence that clusters in the MOTUs, but didn’t amplify. In this case, being able to detect it by backtracking (species specific primers or probes) could be an interesting addition to the study that strenghthens the temporal component.

Additional comments

Study is very relevant and interesting. Identification of basal communities based on genetics and how it can be applied to metabarcoding surveys is a major for experimental design. It's the highest point of the study.

---

## Round 0.2 · accepted · Accept

Both reviewers thought that the revisions were sufficient, but please note that reviewer 2 included some additional feedback.

·

Basic reporting

This manuscript meets all the requirements for basic reporting

Experimental design

This manuscript meets all requirements for experimental design.

Validity of the findings

The authors have improved their discussion of the findings. The manuscript now meets the requirements.

Additional comments

This paper is a very interesting and well-written contribution to the growing field of eDNA metabarcoding. All of my previous concerns have been satisfactorily addressed in your rebuttal and resubmission.

·

Basic reporting

1.1
>English is clear and sentences are well constructed.

1.2
>Context and motivation are well referenced. Methods are clear and satisfactory after revision.

1.3
>Structure is cohesive. Figures and tables are in adequate format. Supplementar material is complete. GenBank sequences are deposited.

1.4
>Figures provided are in good resolution and show what is proposed by the text and analysis

Experimental design

2.1
>Research subject is within biological sciences (molecular ecology).

2.2
>Main question of the study (time correlation) is well defined and relevant. Study answers the question satisfactorily.

2.3
>Experimental design is much clearer after revision.

2.4
>Methods are complete after revision.

A few notes:

1) I asked about the water flow because I didn’t find any info on the water dynamics of the lakes. I was also curious because it is a factor that can induce methodogical bias. Since there seems to be no DNA carryover and it wasn’t controlled, it should not be a problem for the manuscript.

2) I originally asked for DNA concentration because some samples can swamp and inhibit PCR. I was a little bit worried because of the clogging increasing the DNA concentration by a major amount, but after revision, doesn’t seem a problem.

3) PCR plate design doesn’t seem to be a problem with the information provided. I was worried with the cross contamination methodogical bias. Results are very interesting, and it could be undermined by uncertainty introduced by amplicons jumping around.
4) Filtering and chimera control are also ok.

5) By PCR duplicates, I meant the overamplification of a haplotype by random. In a quantification context, It’s important to track this kind of interaction. Since the primer design didn’t include UMIs, it’s very hard to analyze. PCR replicates should be enough to control for this effect then, since it isn’t a absolute quantification study.

6) My mistake on the alpha for UNOISE3. I inverted a parameter when calculating this, so the threshold (B) should indeed decrease very quickly with a 0.5 alpha given d differences between sequences, leading to less MOTUs.

7) Not entirely convinced on the cross-contamination... But since it seems to be an internal issue, instead of a methodological bias, it shouldn’t be a problem. It isn’t clear in the main text though.

8) No problem with the slippages then. I’m not the editor, but I think it’s very interesting to explore the pseudogenes. Maybe not in this paper, but it has a lot of potential for another study if the authors are interested in such kind of analysis.

9) Since pore size was the same for all the samples, temporal correlation should not be affected. It was more of a methodological question, as some studies suggest that composion and abundance can change with the pore size.

10) Extraction replicates not being as impactful as amplification and sampling replicates is very interesting! Since it was tested, should not be a problem too.

Validity of the findings

3.1
>Study shows 2 contributions to metabarcoding and environmental DNA area:
1) Temporal replicates are shown to be an important part of a metabarcoding surveilance design
2) Communities have basal MOTUs that are constant and can be used as control for other experiments, or as biological indicators

3.2
>Data noise and possible biases are explained after revision.

3.3
>Study is limited to the proposed question and answers it satisfactorily. Text is well written and discussion is well fundamented.

3.4
>The following topics discussed in the first revision and this one seem very promissor for future studies:

- Effect of clustering method on the replicate strategies
- Effect of DNA extraction replicates
- Pseudogenes in metabarcoding
- Backtracking MOTUs “jumping” betweeen weeks

While it may not belong to the scope of this study, it certainly can lead to 2-3 more papers for the authors, as data is already available.

Additional comments

Since the methodological biases don't seem to be a problem, basal communities for surveying seems even more promising.